# Let-7c/RUNX1 Axis Promotes Cervical Cancer: A Bioinformatic Analysis

**DOI:** 10.3390/cimb47090757

**Published:** 2025-09-13

**Authors:** Ana Elvira Zacapala-Gómez, Gabriela Hernández-Galicia, Francisco Israel Torres-Rojas, Christian Johana Baños-Hernández, Julio Ortiz-Ortiz, Hilda Jiménez-Wences, Gabriela Elizabeth Campos-Viguri, Verónica Antonio-Véjar, Judit Alarcón-Millán, Eric Genaro Salmerón-Bárcenas

**Affiliations:** 1Laboratorio de Investigación en Bioactivos y Cáncer, Facultad de Ciencias Químico-Biológicas, Universidad Autónoma de Guerrero, Chilpancingo C.P. 39090, Mexico; 17757@uagro.mx; 2Departamento de Biomedicina Molecular, Centro de Investigación y de Estudios Avanzados del Instituto Politécnico Nacional, Ciudad de Mexico City C.P. 07360, Mexico; gabriela.hernandez@cinvestav.mx; 3Laboratorio de Biomedicina Molecular, Facultad de Ciencias Químico-Biológicas, Universidad Autónoma de Guerrero, Chilpancingo C.P. 39090, Mexico; ftorres@uagro.mx; 4Instituto de Investigación en Ciencias Biomédicas, Centro Universitario de Ciencias de la Salud, Universidad de Guadalajara, Guadalajara C.P. 44340, Mexico; johana.banos@academicos.udg.mx; 5Laboratorio de Investigación en Metabolismo y Cáncer, Facultad de Ciencias Químico-Biológicas, Universidad Autónoma de Guerrero, Chilpancingo C.P. 39090, Mexico; julioortiz@uagro.mx (J.O.-O.); hjimenez@uagro.mx (H.J.-W.); 6Laboratorio de Investigación en Biomoléculas, Facultad de Ciencias Químico-Biológicas, Universidad Autónoma de Guerrero, Chilpancingo C.P. 39090, Mexico; 7Laboratorio de Investigación en Enfermedades Infecciosas y Cáncer, Facultad de Ciencias Químico-Biológicas, Universidad Autónoma de Guerrero, Chilpancingo C.P. 39090, Mexico; 8Centro de Investigación en Ciencia Aplicada y Tecnología Avanzada Unidad Morelos, Instituto Politécnico Nacional, Atlacholoaya C.P. 62790, Mexico; gaby_gecv@hotmail.com; 9Laboratorio de Virología y Patología Traslacional, Facultad de Ciencias Químico-Biológicas, Universidad Autónoma de Guerrero, Chilpancingo C.P. 39090, Mexico; 11335@uagro.mx; 10Laboratorio de Investigación en Bacteriología, Facultad de Ciencias Químico-Biológicas, Universidad Autónoma de Guerrero, Chilpancingo C.P. 39090, Mexico

**Keywords:** Let-7c, RUNX1, cervical cancer, bioinformatic

## Abstract

Background: Cervical cancer (CC) ranks as the third most common cancer in incidence and mortality in females worldwide. Let-7c is a tumor suppressor miRNA, and its role has been little studied in CC. Runt-related transcription factor 1 (RUNX1) is upregulated in several human cancers, such as colorectal cancer. It is a transcription factor that promotes cell proliferation, metastasis, chemotherapy resistance and angiogenesis in colorectal cancer. In this study, we performed a bioinformatic analysis to understand how Let-7c and RUNX1 are involved in the development of CC. Methods: We performed a bioinformatic analysis of Let-7c in CC using GSE and TCGA datasets from GEO, KM-plotter, miRPathDB and Enrich databases. Then, we conducted a comprehensive analysis of RUNX1’s role in CC using TCGA, GSE and HPA datasets from OncoDB, CISTROME, ExPASy, Alibaba, ALGGEN, ENCODE, IGV, GEO, KM-plotter and DiseaseMeth databases. Results: We found that Let-7c expression is decreased in CC. Interestingly, we identified a transcription factor known as RUNX1, as a potential target of Let-7c. Finally, we suggest that RUNX1 could regulate the expression of several genes, promoting CC. Conclusions: The Let-7c/RUNX1 axis promotes CC.

## 1. Introduction

Cervical cancer (CC) occupies the third place in incidence and mortality among females worldwide, with a total of 660,000 new cases and 350,000 deaths estimated in 2022 [1]. Recently, 14,000 new cases and 4280 deaths were reported in the United States in 2022 [2]. It is well known that persistent infection with High-Risk Human Papillomavirus (HR-HPV) is a key risk factor for CC development, but it is not the only one [3]. In this sense, several key non-coding RNAs have been identified, including lncRNAs, circRNAs and miRNAs [4].

miRNAs are small non-coding RNAs of ~22 nucleotides that inhibit the expression of target genes at the post-transcriptional level by binding to a specific sequence known as “seed” located in 3′ UTR, 5′ UTR or coding regions of target genes. A total of 2654 miRNAs have been identified in humans, and their deregulation alters key cell processes, such as cell migration, invasion and cell cycle. Therefore, miRNAs play a very important role in human disease, including CC [5,6]. Recently, it was reported that Let-7c overexpression inhibits cell adhesion, invasion and migration in CC by downregulation of CHD7 [7]. However, its role remains largely unknown in CC.

Runt-related transcription factor 1 (RUNX1) is a transcription factor that regulates the fate of stem cells and participates in the development of the central and peripheral nervous systems [8,9]. RUNX1 expression is increased in several human cancers, such as colorectal cancer [9]. In some human cancers, the role of RUNX1 is well known; for example, RUNX1 promotes cell proliferation, metastasis, chemotherapy resistance and angiogenesis in colorectal cancer cell lines [10]. Similarly, RUNX1 promotes cell proliferation and migration in non-small-cell lung cancer cell lines through hyperactivation of the mTORC pathway [11]. However, the role of RUNX1 in CC is not fully understood.

In this study, we performed a bioinformatic analysis of Let-7c in CC using GSE and TCGA datasets from GEO, KM-plotter, miRPathDB and Enrich databases. We found that Let-7c expression is decreased in CC. Interestingly, we identified a transcription factor, RUNX1, as a potential target of Let-7c using the miRPathDB database. Then, we performed a comprehensive analysis of RUNX1’s role in CC using TCGA, GSE and HPA datasets from OncoDB, GEO, KM-plotter, DiseaseMeth and CISTROME databases. Finally, our results suggest that RUNX1 could promote the expression of oncogenes such as FOXO34 and CDC42BPG, as well as inhibit the expression of tumor suppressor genes such as FBN1 and SFRP4.

## 2. Materials and Methods

### 2.1. Expression Analysis

The expression analysis was performed in normal and CC tissue samples from TCGA, GSE30656 [12], GSE86100 [13], GSE67522 [14], GSE63514 [15] and Human Protein Atlas (HPA) [16] datasets using oncoMir Cancer [17], OncoDB [18], GEPIA [19] and Gene Expression Omnibus (GEO) [20] databases (Table 1). The data were Log2-transformed and are shown as median ± SD. The differences were determined with Student’s *t*-test and an ANOVA test (only data from GEPIA). A *p*-value < 0.05 was considered significant.

### 2.2. ROC Curve Analysis

The analysis of the Receiver Operating Characteristic (ROC) curve was performed using GraphPad Prism version 8.0.1 (for Windows, GraphPad Software, San Diego, CA, USA, www.graphpad.com) (accessed on 23 September 2023). The Area Under the Curve (AUC) was calculated considering a 95% Confidence Interval (CI). A *p*-value ˂ 0.05 was considered significant, and an AUC ≥ 7 was considered acceptable [21]. In addition, the values of optimal cut-off, sensitivity and specificity were determined using the Youden index method [22].

### 2.3. Overall Survival Analysis

Overall Survival (OS) and Relapse-Free Survival (RFS) analyses were performed through Kaplan–Meier curves in CC tissue samples from the TCGA dataset using the KM-plotter database [23]. The median of expression of each transcript was used as cut-off. The Log-Rank and Hazard Ratio tests were performed, and a *p*-value ˂ 0.05 was considered as statistically significant.

### 2.4. Analysis of Target Identification

The identification of target mRNAs for miR-Let-7c was performed in the miRPathDB V2.0 database [24]. All targets were selected, including both predicted and experimentally validated. For RUNX1, the target genes were identified using the CISTROME database [25]. The selection criteria were the expression correlation and regular potential score cut-off (%) of target genes. The common targets for RUNX1 were identified using the Venn diagram online tool (https://bioinformatics.psb.ugent.be/webtools/Venn/) (accessed on 14 October 2024).

### 2.5. Pathway and Biological Process (BP) Analysis

Analysis of pathways and BPs was performed using the Bioplanet 2019, MSigDB Hallmark 2020, KEGG 2021 Human and Biological Process 2021 libraries in the Enrich database [26]. The genes in common for top pathways and BPs were identified using the Venn diagram online tool (https://bioinformatics.psb.ugent.be/webtools/Venn/) (accessed on 12 March 2024). A *p*-value ˂ 0.05 was significant.

### 2.6. CpG Island Prediction on RUNX1 Promoter

The *RUNX1* promoter was downloaded from the Eukaryotic Promoter Database (EPD) [27] from the Expert Protein Analysis System (ExPASy) portal (https://www.expasy.org/) (accessed on 29 June 2024) [28]. The DNA sequence 4000 pb to +4000 pb was selected for CpG island prediction, which was performed using the Meth-Primer program [29] with the following parameters: window size of 200, shift of 1, Obs/Exp of 0.6 and GC% of 50.

### 2.7. Methylation Analysis

The methylation analysis was performed in normal and CC tissue samples from TCGA and GSE30760 [30] datasets using DiseaseMeth V3.0 [31] and GEO [20] databases (Table 1). The data is shown as median ± SD, and the differences were determined using Student’s *t*-test. A *p*-value < 0.05 was considered significant.

### 2.8. Transcription Factor Identification

The transcription factors binding to the *RUNX1* promoter were identified using the Alibaba v2.1 [32] and ALGGEN [33] databases with the default parameters. Common transcription factors were identified using the Venn diagram online tool (https://bioinformatics.psb.ugent.be/webtools/Venn/) (accessed on 4 July 2024). The direct binding of the AP-2α transcription factor to the *RUNX1* promoter was analyzed using data from the ENCODE project (2012) (accessed on 2 October 2024) and visualized in Integrative Genomics Viewer (IGV) v2.12.3 software [34].

## 3. Results

### 3.1. miR-Let-7c Expression Decreases in CC

To determine the miR-Let-7c expression in CC, we analyzed its expression in CC and normal tissue samples from TCGA, GSE30656 and GSE86100 datasets using the OncoMir Cancer and GEO databases. The results revealed that miR-Let-7c expression decreases in CC tissue samples (Figure 1A–C).

To analyze the diagnostic and prognostic value of miR-Let-7c expression in this type of cancer, we performed ROC and Kaplan–Meier curves considering the expression of this miRNA in CC and normal tissue samples from GSE30656 and TCGA datasets using GraphPad Prism software and the KM-plotter database. We observed an AUC of 0.7842 (*p*-value: 0.0132), with a sensitivity and specificity of 78.95% and 80.00% (cut-off: 8.990), respectively (Figure 1D). Moreover, we found that low miR-Let-7c expression correlates with poor OS in patients with this type of cancer (Figure 1E).

Altogether, our results suggest that miR-Let-7c expression decreases in CC and could be useful as a diagnostic and prognostic biomarker in patients with CC.

### 3.2. Identification of Pathways and BPs Associated with Let-7c Targets in CC

To investigate the role of Let-7c in CC, we searched its potential targets in CC using the miRPathDB database. A total of 6888 potential targets were identified. Next, we performed pathway and BP analysis using MSigDB Hallmark 2020 and GO Biological Processes 2023 libraries in the Enrich catalog. We identified key signaling pathways in CC, such as UV response dn, IL-2/STAT5 signaling, the p53 pathway, mTOC1 signaling and PI3K/AKT/mTOR signaling (Figure 2A). Moreover, the BPs identified were Regulation of DNA-templated transcription, Regulation of MAPK cascade, Regulation of transcription by RNA polymerase II, Phosphorylation and Negative regulation of gene expression (Figure 2B). Finally, a Venn diagram between the target mRNAs involved in the top signaling pathway (UV response dn) and the top BP (Regulation of DNA-templated transcription) revealed 16 potential targets in common, including ID1, MYC and RUNX1 (Figure 2C).

These results suggest that the Let-7c downregulation could be involved in cervical carcinogenesis through the upregulation of its target genes that participate in key events in CC.

### 3.3. RUNX1 Expression Increases in CC

Recently, it was reported that RUNX1 expression is associated with immune infiltrates of cancer-associated fibroblasts in several types of cancers, including cervical [9]; however, its role has not been fully studied in CC, and therefore, we focus on this gene.

To determine the RUNX1 expression in CC, we analyzed its expression in CC and normal tissue samples from TCGA and GSE67522 datasets using OncoDB and GEO databases. The results revealed that RUNX1 expression increases in CC tissue samples (Figure 3A,B). To confirm these results, we analyzed the RUNX1 expression at the protein level in CC and normal tissue samples from the HPA database, and similar results were found (Figure 3C).

To explore the diagnostic and prognostic utility of the RUNX1 expression in CC, an analysis of ROC and Kaplan–Meier curves was performed according to the RUNX1 expression in CC and normal tissue samples from GSE67522 and TCGA datasets in the GraphPad Prism software program and KM-plotter database. The ROC curve analysis showed an AUC of 0.8023 (*p*-value: 0.0008), with a sensitivity of 75.00% and a specificity of 77.27% (cut-off: 7.901) (Figure 3D). In addition, we found that high RUX1 expression is associated with poor OS (Figure 3E) and RFS (Figure 3F) in CC patients.

These results suggest that RUNX1 expression increases in CC and could be a diagnostic and prognostic biomarker.

### 3.4. Methylation Levels Increase in RUNX1 Promoter in CC

To explore whether methylation promotes the high RUNX1 expression in CC, we searched for the presence of CpG islands in the *RUNX1* promoter using the Methprimer program. We identified a CpG island located specifically between −3600 pb and +400 pb (Figure 4A). Next, we analyzed the methylation level in the *RUNX1* promoter in CC and normal tissue samples from TCGA and GSE30760 datasets using DiseaseMeth and GEO databases. Surprisingly, we found that methylation level increases in CC tissue samples (Figure 4B–D).

Our findings suggest that aberrant methylation on the *RUNX1* promoter could induce the recruitment of oncogenic transcription factors, increasing the RUNX1 expression in CC.

### 3.5. The AP-2α Transcription Factor Binds to RUNX1 Promoter in HeLa-S3 CC Cell Line

Several studies have shown that hypermethylation in gene promoters induces the binding of transcription factors, such as NFATc1 [35,36,37]. Therefore, we searched for transcription factors that could bind to the *RUNX1* promoter (specifically in the region analyzed in Figure 4C,D) using Alibaba and ALLGENE databases. As shown in Figure 5A,B, we found binding sites of four transcription factors (C/EBPα, C/EBPβ, NF-1 and AP-2α) that could bind to the previously analyzed regions. Previous studies have identified C/EBPα [38], C/EBPβ [39] and NF-1 [40] as tumor suppressors, whereas AP-2α acts as an oncogene [41,42]. Therefore, we analyzed the expression of this transcription factor in CC and normal tissue samples from the TCGA dataset using the GEPIA platform. We observed that AP-2α levels are increased in this type of cancer (Figure 5C).

To evaluate whether AP-2α directly induces the RUNX1 expression, we analyzed the binding of this transcription factor to the *RUNX1* promoter using ChIP-seq assays in the HeLa-S3 CC cell line from the IGV database. Interestingly, we identified three binding sites of AP-2α on the *RUNX1* gene, specifically one binding site is located in the RUNX1 promoter (Red box) (Figure 5D).

Overall, these results suggest that the AP-2α transcription factor could be recruited to the *RUNX1* promoter via aberrant methylation and induce its overexpression in CC.

### 3.6. Identification of Pathways Regulated Through RUNX1 in CC

To investigate the potential pathways regulated by RUNX1 in CC, we searched target genes considering the expression correlation and regulatory potential of target genes in the CISTROME database. We identified a total of 52 and 3325 targets. Surprisingly, a Venn diagram revealed a total of 52 common target genes (Figure 6A). Next, we performed a BP and pathway analysis using GO Biological process 2023, KEGG 2021 Human and Bioplanet 2019 libraries in the Enrich database. As Figure 6B shows, we identified key BPs to cervical carcinogenesis, such as Cellular response to TGFβ stimulus, Response to TGFβ, Cell–Matrix adhesion and Regulation of the non-canonical Wnt signaling pathway. Moreover, the pathway analysis in the KEGG 2021 Human library revealed key pathways in CC, including the Wnt signaling pathway, cGMP-PKG signaling pathway and Pathways in cancer (Figure 6C). Similar results were obtained using the BioPlanet 2019 library, including TGF-beta regulation of the extracellular matrix, the Wnt signaling pathway and the BDNF signaling pathway (Figure 6D).

Altogether, these results suggest that the RUNX1 expression could promote CC via deregulation of expression of genes involved in key signaling pathways to CC, such as the Wnt signaling pathway, cGMP-PKG signaling pathway and Pathways in cancer.

### 3.7. Analysis of Expression of 4 Potential Target Genes of RUNX1 in CC

To determine the role of RUNX1 in the expression of its potential target genes, we analyzed the expression of four randomly selected potential target genes in CC and normal tissue samples from the TCGA dataset using the OncoDB database. Interestingly, we found that FBXO34 and CDC42BPG expression is increased in CC samples (Figure 7A,B), while SFRP4 and FBN1 expression is decreased in CC samples (Figure 7C,D).

These results suggest that the FBXO34, CDC42BPG, SFRP4 and FBN1 expression could be partially altered by RUNX1 in CC.

### 3.8. Prognostic Value of Four Potential Target Genes of RUNX1 in CC

To explore the potential prognostic value of FOXO34, CDC42BPG, FBN1 and SFRP4 in CC, we performed a Kaplan–Meier curve analysis considering the individual expression of each potential target in CC tissue samples from the TCGA dataset in the KM-plotter database. The results revealed that high FOXO34 and CDC42BPG expression does not correlate with poor OS in patients with CC (Figure 8A,B). However, high FBN1 expression correlates with poor OS (Figure 8C). Consistently, we found that low SFRP4 expression correlates with poor OS in CC patients (Figure 8D).

These results suggest that the low SFRP4 expression could be useful as a prognostic biomarker in CC.

## 4. Discussion

CC is a public health problem worldwide. According to GLOBOCAN, CC is third at both mortality and incidence among female cancers [1]. Therefore, it is required to elucidate the molecular mechanisms involved in its origin and development to propose new diagnostic, prognostic and therapeutic biomarkers. In this context, several studies have shown that miRNAs are key molecules in the development of CC [43], representing new opportunities to reduce the high mortality and incidence rates of this cancer.

Recently, a study showed that Let-7c-5p overexpression decreases the migration, invasion and adhesion by CHD7 downregulation in HeLa CC cells; however, the role of Let-7c-5p was not analyzed in depth. In the present study, the Let-7c expression was analyzed in tissue samples of CC patients of three datasets from independent cohorts, and consistently, the results showed low Let-7c expression in CC, suggesting a potential role as a tumor suppressor miRNA. Therefore, it will be very interesting to analyze the molecular mechanisms involved in Let-7c downregulation in CC, such as transcription factors, lncRNAs, DNA methylation and circRNAs, as well as HR-HPV (oncogenic genotypes). Moreover, the potential value of low Let-7c expression in CC as a diagnostic and prognostic biomarker was explored in our study, and these results suggest that low Let-7c expression could be used as a biomarker; however, it is necessary to confirm the results using a larger dataset.

In addition, the identification of potential targets of Let-7c, as well as the analysis of pathways, strongly suggests that low Let-7c expression could increase the expression levels of genes involved in key pathways to CC, such as p53 and mTORC1 signaling pathways, suggesting a potential role of tumor suppressor miRNA for Let-7c in CC. A detailed analysis of pathways and BPs revealed 16 potential targets and key genes in development of CC via Let-7c. These results are consistent with previous studies. For example, ID1 overexpression positively correlates with tumor growth, invasion and metastasis in patients with CC [44]. The Myc gene was identified as a potential target of Let-7c, and a previous study reported its role as an oncogene in CC patients [45]. Similarly, COL1A1 is overexpressed in CC and inhibits the apoptosis induced by radiation in CaSKi and HeLa CC cells [46]. Therefore, these studies support our results and validate our analysis.

On the other hand, in this work we focused on the transcription factor RUNX1 because the dysregulation of transcription factors is a key event in cancer [47]. Moreover, the role of RUNX1 is not yet fully understood in CC. Our results suggest that RUNX1 expression increases in CC, suggesting that it could be useful as a diagnostic biomarker. In this sense, a previous study revealed that RUNX1 expression increases in patients with CC and induces migration and invasion in CaSKi CC cells [48]. In addition, during the preparation of our manuscript, a study was published and revealed that RUNX1 promotes cell proliferation and in vivo growth through MAPK pathway overactivation via TGFB2 upregulation [49]. However, these studies did not investigate the molecular mechanism involved in RUNX1 overexpression in CC. In our work, we analyzed the potential molecular mechanisms that could be involved in RUNX1 overexpression. Surprisingly, we found high levels of DNA methylation in the *RUNX1* promoter. Currently, there is evidence that shows that promoter hypermethylation promotes high levels of gene expression in cancer by recruitment of oncogenic transcription factors [36,50,51]. Interestingly, we identified that the transcription factor AP-2α binds to the *RUNX1* promoter in HeLa-S3 CC cells. A study showed that AP-2α is an oncogenic transcription factor that increases the PD-L1 expression, promoting cell proliferation, migration and invasion and inhibiting apoptosis in HeLa and SiHa CC cells [42]. Therefore, our results contribute to clarifying the molecular mechanism that promotes RUNX1 overexpression and suggest that the AP-2α/RUNX1 axis is a key event in CC. However, we do not rule out the possibility of participation of other transcription factors, the alterations in the copy number and other miRNAs that promote RUNX1 overexpression in this disease.

To understand the role of RUNX1 in CC, we conducted a robust workflow, identifying its targets from integrative analysis of TCGA expression profiles and public ChIP-seq profiles in CC. Interestingly, these results revealed that RUNX1 could have 52 direct targets in CC. In contrast, a recent study identified a total of 2534 altered genes in CaSki CC cells with RUNX1 knockdown. However, these results were only obtained through RNA-seq assays, and therefore, these genes could be direct or indirect targets of RUNX1. Moreover, these results are limited to only a single cell line, which is a cell line positive for HPV16 [49]. Consistently, our pathway analysis revealed that target genes of RUNX1 participate in key pathways, which have been previously reported in this type of cancer, such as Pathways in cancer [52], WNT signaling pathways [53] and the BDNF signaling pathway [54]. Moreover, we validate the expression of four potential target genes of RUNX1 in CC, FBXO34, CDC42BPG, FBN1 and SFRP4, and their expression was altered in CC, suggesting that RUNX1 could be a potential regulator of these genes in CC. In addition, previous studies have reported that FBN1 [55] and SFRP4 [56] act as tumor suppressor genes, supporting our results.

Our work has some limitations. First, all our results need to be validated experimentally, such as knockdown, rescue and RT-qPCR assays. Second, it is necessary to increase the number of samples in some analyses, such as the analysis of expression in the TCGA dataset, as well as the ROC analysis in the datasets. However, our results were consistent in several datasets analyzed, significantly supporting our findings.

## 5. Conclusions

We found low Let-7c expression in CC, which could induce RUNX1 overexpression, an oncogenic transcription factor, promoting the alteration of gene expression that promotes CC development. Moreover, Let-7c and RUNX1 expression could be considered as a potential diagnostic and prognostic biomarker. Finally, our results suggest that the Let-7c/RUNX1 axis is key in CC.

## Figures and Tables

**Figure 1 cimb-47-00757-f001:**
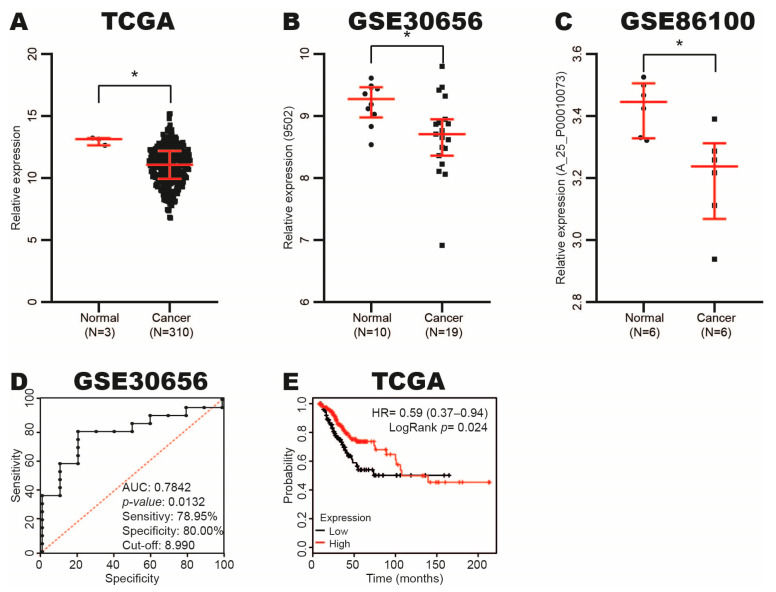
Expression and diagnostic and prognostic value of miR-Let-7c in CC. (**A**–**C**) miR-Let-7c expression in CC and normal tissue samples from TCGA, GSE30656 and GSE86100 datasets using miR-TV and GEO databases, respectively. (**D**) Diagnostic value of miR-Let-7c expression in CC and normal tissue samples from GSE30656 dataset analyzed with ROC curve using GraphPad Prism software. (**E**) Prognostic value of miR-Let-7c expression in CC tissue samples from TCGA dataset analyzed by Kaplan–Meier curve using KM-plotter. * *p*-value ˂ 0.05; black dots: normal cervical samples; black squares: CC samples.

**Figure 2 cimb-47-00757-f002:**
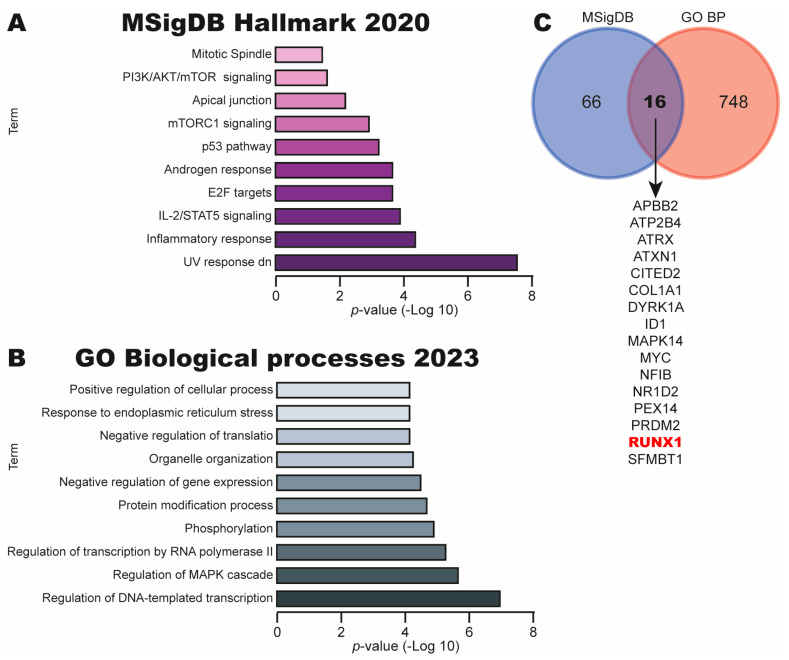
Identification of pathways and BPs associated with target mRNAs of miR-Let-7c in CC. (**A**) Top ten pathways identified using MSigDB Hallmark 2020 in Enrich database. (**B**) Top ten BPs using GO Biological process 2023 library in Enrich catalog. (**C**) Venn diagram of common genes from top pathways and top BPs.

**Figure 3 cimb-47-00757-f003:**
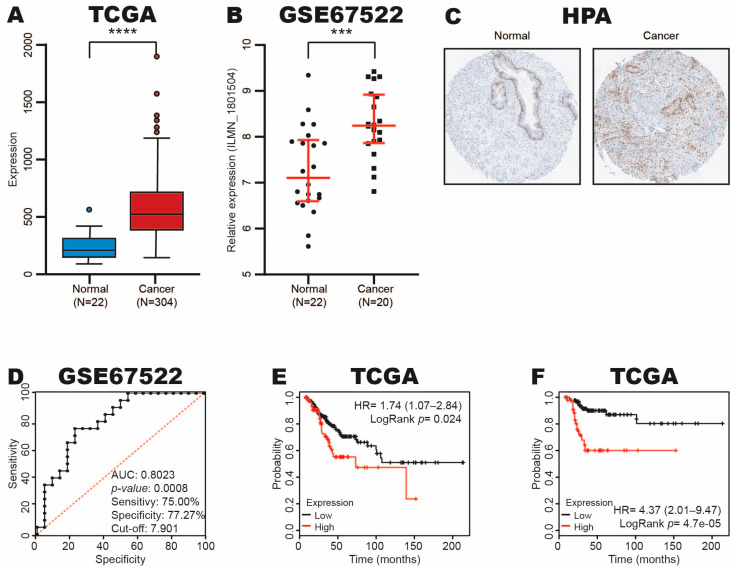
Expression, diagnostic and prognostic value of RUNX1 in CC. (**A**–**C**) RUNX1 expression in CC and normal samples from TCGA, GSE67522 and HPA datasets using OncoDB, GEO and HPA databases, respectively. (**D**) Diagnostic value of RUNX1 expression in CC and control samples from GSE67522 dataset analyzed by ROC curves in GraphPad Prism software. (**E**,**F**). Prognostic value of RUNX1 expression in CC tissue samples from TCGA dataset analyzed with Kaplan–Meier curves. *** *p*-value ˂ 0.001; **** *p*-value ˂ 0.0001; black dots: normal cervical samples; black squares: CC samples.

**Figure 4 cimb-47-00757-f004:**
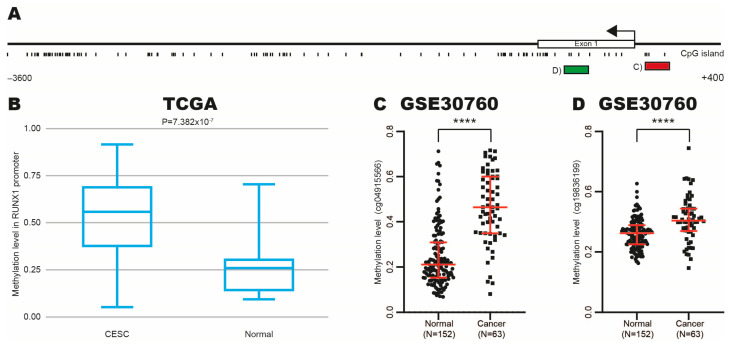
Identification of methylation levels on *RUNX1* promoter in CC. (**A**) A CpG island in *RUNX1* promoter using MethPrimer program. (**B**–**D**) Methylation level at *RUNX1* promoter in CC and normal samples from TCGA and GSE30760 datasets with DiseaseMeth and GEO databases. (**B**) Entire promoter was analyzed. (**C**) Region analyzed is represented in red box in A. (**D**) Region analyzed is represented by green box in A. **** *p*-value ˂ 0.0001; black dots: normal cervical samples; black squares: CC samples.

**Figure 5 cimb-47-00757-f005:**
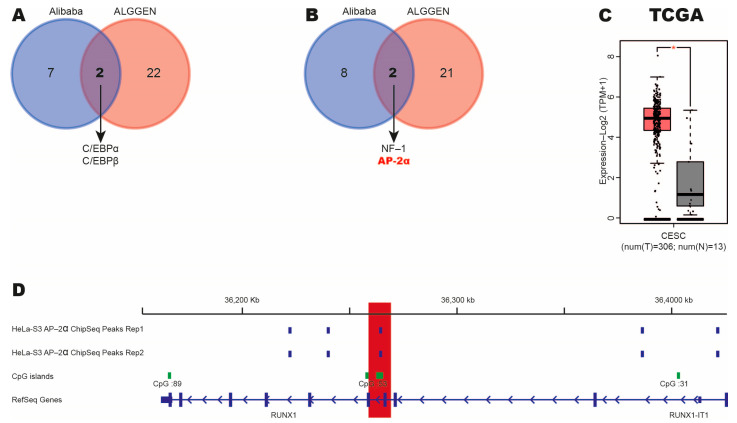
Identification of transcription factors that regulate RUNX1 expression in CC. (**A**,**B**) Venn diagrams of common transcription factors identified in Figure 4C,D using CONSITE and ALLGENE databases. (**C**) AP-2α level in CC and normal samples from TCGA. Red box represent CC samples. Grey box represent normal cervical samples. * *p*-value ˂ 0.05. (**D**) Track showing binding of AP-2α at *RUNX1* promoter in HeLa-S3 CC cell line using IGV program. Two biological replicates were analyzed. Green boxes represent CpG islands. Red box indicates binding site of AP-2α on *RUNX1* promoter. Blue boxes represent binding sites on RUNX1.

**Figure 6 cimb-47-00757-f006:**
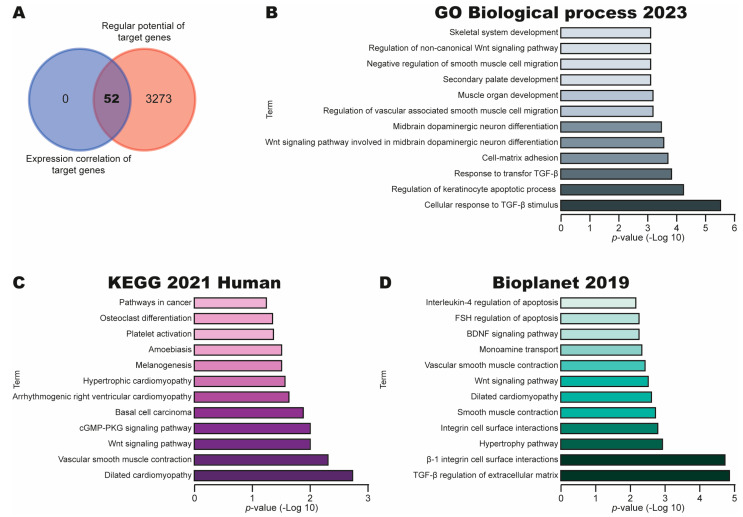
Identification of BPs and pathways associated with target genes of RUNX1 in CC. (**A**) Venn diagram of common target genes of RUNX1 using CISTROME database. (**B**) Top BPs with 52 target genes of RUNX1 using GO Biological processes 2023 library in Enrich database. (**C**,**D**) Top pathways with 52 target genes of RUNX1 using KEGG 2021 Human and Bioplanet 2019 libraries in Enrich database.

**Figure 7 cimb-47-00757-f007:**
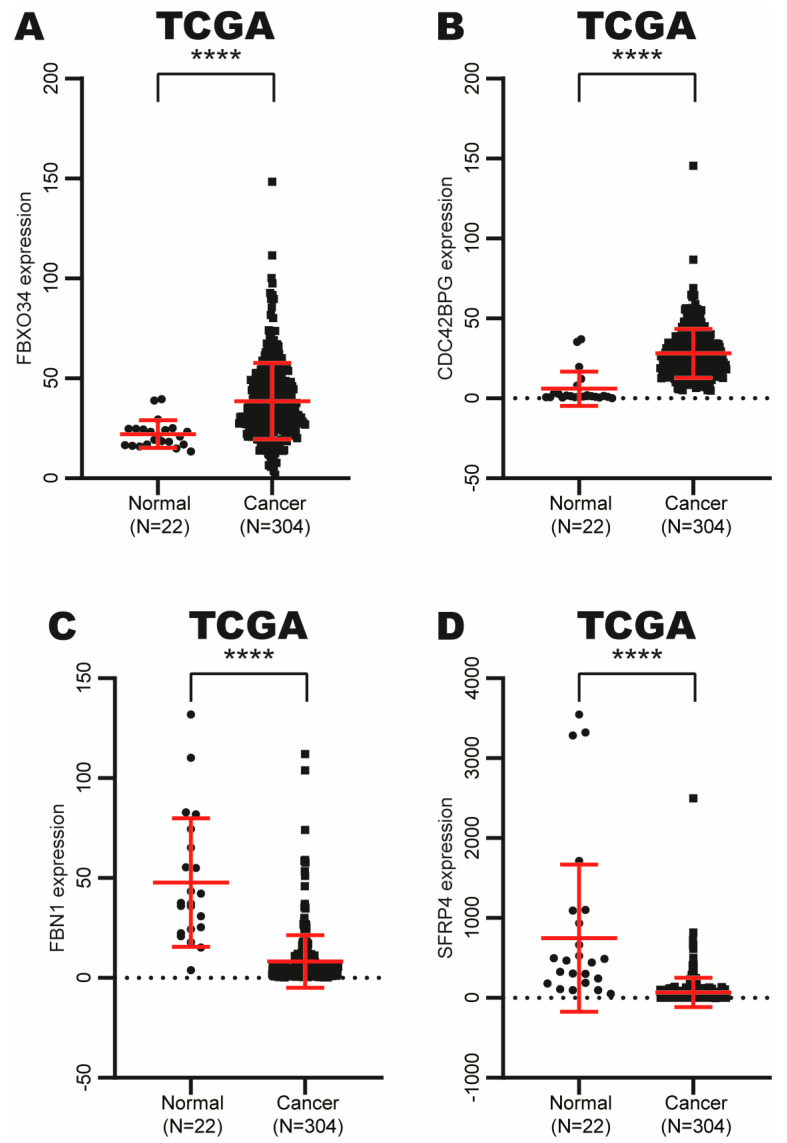
Expression of target genes of RUNX1 in CC. Expression analysis of FOXO34 (**A**), CDC42BPG (**B**), FBN1 (**C**) and SFRP4 (**D**) in CC and normal tissue samples from TCGA dataset using OncoDB database. **** *p*-value ˂ 0.0001; black dots: normal cervical samples; black squares: CC samples.

**Figure 8 cimb-47-00757-f008:**
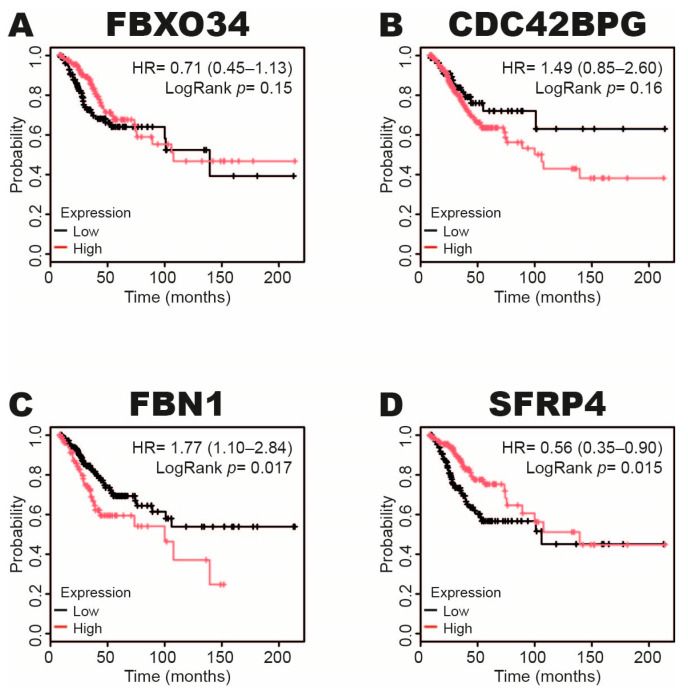
Prognostic value of four potential target genes of RUNX1 in CC. Prognostic values of (**A**) FBXO34, (**B**) CDC42BPG, (**C**) FBN1 and (**D**) SFRP4 expression in patients with CC using KM-plotter database.

**Table 1 cimb-47-00757-t001:** Datasets used in this study.

Dataset	Database	Study	Platform	Total Samples (CC/Controls)	Assay	PMID
TCGA	OncoMir	miRNAs	NA	313 (310/3)	RNA-seq	30522456
GSE30656	GEO	miRNAs	GPL6955	29 (19/10)	Microarray	22330141/11752295
GSE86100	GEO	miRNAs	GPL19730	12 (6/6)	Microarray	27764149/11752295
TCGA	OncoDB	mRNAs	NA	326 (304/22)	RNA-seq	34718715
TCGA	GEPIA	mRNAs	NA	3 19(306/13)	RNA-seq	28407145
GSE67522	GEO	mRNAs	GPL10558	42 (20/22)	Microarray	26152361/11752295
GSE63514	GEO	mRNAs	GPL580	52 (28/24)	Microarray	26056290/11752295
TCGA	DiseaseMeth	Methylation	GPL13534	313 (310/3)	Microarray	34792145
GSE30760	GEO	Methylation	GPL8490	215 (63/152)	Microarray	22453031/11752295

NA: Not applicable.

## Data Availability

These data were derived from the following resources available in the public domain: [GEO] [https://www.ncbi.nlm.nih.gov/geo/] [20]; [TCGA] [http://gepia.cancer-pku.cn/] [19]; [TCGA] [https://oncodb.org/] [18]; [TCGA] [https://oncomir.org/] [17]; [HPA] [https://www.proteinatlas.org/] [16].

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
