# Peer review of "Let-7c/RUNX1 Axis Promotes Cervical Cancer: A Bioinformatic Analysis"

_cimb, 2025, doi:10.3390/cimb47090757_

Round 1
Reviewer 1 Report
Comments and Suggestions for Authors
Major
1) For pathway, GSEA and other statistical tests in method section, shouldn’t the significant value be selected from the adjusted p-value instead of the p-value for multiple testing?
2) Please justify the selection of the sequence -3,600 bp to +400 bp to predict CpG islands on RUNX1 promoter? Why is the selected range adequate to use, and why didn’t the authors use the CpG island information available on the track in ExPASy?
Minor
1) Please check and correct all typos throughout the manuscript, e.g., bp instead of pb, CDC42BPG instead of CDD42BPG, the excessive use of "and" in "Let-7c and RUNX1 and expression" in the conclusions section
2) Please reduce or remove the bold database title in all figures, e.g., TGCA in Figure 7. Describing the use of the database in each figure panel in the figure legend should serve this purpose.
Author Response
|
1. Summary |
|
|
|
Thank you very much for taking the time to review this manuscript. Please find the detailed responses below and the corresponding revisions/corrections highlighted/in track changes in the re-submitted files.
|
||
|
2. Questions for General Evaluation |
Reviewer’s Evaluation |
Response and Revisions |
|
Does the introduction provide sufficient background and include all relevant references? |
Yes/Can be improved/Must be improved/Not applicable |
Not applicable |
|
Are all the cited references relevant to the research? |
Yes/Can be improved/Must be improved/Not applicable |
Cited references were revised. |
|
Is the research design appropriate? |
Yes/Can be improved/Must be improved/Not applicable |
The workflow was revised. |
|
Are the methods adequately described? |
Yes/Can be improved/Must be improved/Not applicable |
Changes are marked in red letters. |
|
Are the results clearly presented? |
Yes/Can be improved/Must be improved/Not applicable |
The results were reviewed. |
|
Are the conclusions supported by the results? |
Yes/Can be improved/Must be improved/Not applicable |
Not applicable |
|
3. Point-by-point response to Comments and Suggestions for Authors |
||
|
Comments 1: For pathway, GSEA and other statistical tests in method section, shouldn’t the significant value be selected from the adjusted p-value instead of the p-value for multiple testing? |
||
|
Response 1: Thank you for pointing this out. We do not agree with this comment given that statistical tests can be p-value [1, 2] or adjusted p-value [3]. In our study, we choose selection based on P-Value Ranking.
|
||
|
Comments 2: Please justify the selection of the sequence -3,600 bp to +400 bp to predict CpG islands on RUNX1 promoter? Why is the selected range adequate to use, and why didn’t the authors use the CpG island information available on the track in ExPASy? |
||
|
Response 2: Agree. We modified the redaction in sections: 2.6. CpG island prediction on RUNX1 promoter, and 3.4. Methylation levels increase in RUNX1 promoter in CC, to emphasize this point. The text updated in the manuscript in: Section 2.6.: The RUNX1 promoter was downloaded from the Eukaryotic Promoter Database (EPD) [27] from the Expert Protein Analysis System (ExPASy) portal [28]. The sequence -4,000 pb to +4000 pb was selected for the CpG island prediction, which was performed using Meth-Primer program [29] considering a window: 200, shift: 1, Obs/Exp: 0.6 and GC%: 50. Section 3.4.: To explore if methylation is involved in RUNX1 overexpression in CC, first we search the presence of CpG islands in the RUNX1 promoter using the Methprimer program and we found a CpG island located specifically between 3,600 pb to +400 pb (Figure 4A).
Comments 3: Please check and correct all typos throughout the manuscript, e.g., bp instead of pb, CDC42BPG instead of CDD42BPG, the excessive use of "and" in "Let-7c and RUNX1 and expression" in the conclusions section Response 3: Agree. We modified the redaction in new version of our manuscript to emphasize this point. For example, CDC42BPG was corrected in Discussion.
Comments 4: Please reduce or remove the bold database title in all figures, e.g., TGCA in Figure 7. Describing the use of the database in each figure panel in the figure legend should serve this purpose. Response 4: Agree. We modified all figures in the new version of our manuscript. |
||
|
4. Response to Comments on the Quality of English Language |
||
|
Point 1: |
||
|
Response 1: The Quality of English Language in current version of our manuscript was improved by American professional Scientifics (in red). |
||
|
5. Additional clarifications |
||
|
The current version of our manuscript was critically revised and evaluated by several professional Scientifics. Changes are marked in red letters. Figure quality was improved. |
||
Reviewer 2 Report
Comments and Suggestions for Authors
Please review the attached file.

Comments on the Quality of English Language
This manuscript contains valuable scientific content, and the research is well designed. However, frequent errors in English grammar, syntax, and phrasing currently diminish its clarity and overall impact.
To improve readability and ensure the manuscript meets the journal's publication standards, a thorough revision by a native English speaker or professional scientific editor is strongly recommended. Resolving these language issues will enable you to communicate your important findings clearly and effectively to the scientific community.
Author Response
|
1. Summary |
|
|
|
Thank you very much for taking the time to review this manuscript. Please find the detailed responses below and the corresponding revisions/corrections highlighted/in track changes in the re-submitted files.
|
||
|
2. Questions for General Evaluation |
Reviewer’s Evaluation |
Response and Revisions |
|
Does the introduction provide sufficient background and include all relevant references? |
Yes/Can be improved/Must be improved/Not applicable |
Thanks. |
|
Are all the cited references relevant to the research? |
Yes/Can be improved/Must be improved/Not applicable |
Cited references were reviewed. |
|
Is the research design appropriate? |
Yes/Can be improved/Must be improved/Not applicable |
The workflow was update. |
|
Are the methods adequately described? |
Yes/Can be improved/Must be improved/Not applicable |
Changes are marked in red letters. |
|
Are the results clearly presented? |
Yes/Can be improved/Must be improved/Not applicable |
The results were reviewed. |
|
Are the conclusions supported by the results? |
Yes/Can be improved/Must be improved/Not applicable |
Not applicable |
|
3. Point-by-point response to Comments and Suggestions for Authors |
||
|
Comments 1: This manuscript contains valuable scientific content, and the research is well designed. However, frequent errors in English grammar, syntax, and phrasing currently diminish its clarity and overall impact. To improve readability and ensure the manuscript meets the journal's publication standards, a thorough revision by a native English speaker or professional scientific editor is strongly recommended. Resolving these language issues will enable you to communicate your important findings clearly and effectively to the scientific community. |
||
|
Response 1: Thank you for pointing this out. We agree with this comment. English grammar and syntax were revised by professional Scientifics. The manuscript was updated, and the changes are marked in red letters.
Comments 2: The primary area requiring attention is English grammar and usage. While the manuscript's scientific content is strong, it could benefit from a thorough review by a native English speaker or professional editing service. Improving the language can enhance the clarity, flow, and overall impact of your important findings. Below are some examples of sentences that could be improved for clarity and grammatical correctness: Abstract and Introduction: The phrase "is at the top in terms of incidence and mortality" is slightly awkward. It could be rephrasing similar to "It is one of the most common cancers in terms of incidence and mortality." ? Introduction: The sentence "Key non-coding RNAs have been identified". It could be "Several key non-coding RNAs have been identified. "Methods: In Section 2.2, "An AUC of 27 was considered acceptable" appears to be a typo. It should be corrected to "an AUC >0.7." Response 2: Thank you for pointing this out. We agree with your comments. English grammar and syntax were revised by professional Scientifics. In our new version, the changes are marked in red letters.
Comments 3: Please proofread the manuscript carefully to correct any minor typos. For instance: • In the Results section discussing Figure 3, "high RUX1 expression" is mentioned. This should be corrected to "RUNX1." • The legend for Figure 8B refers to "CDD42BPG," but the text and figure title use the correct term, "CDC42BPG." Please ensure consistency Response 3: Thank you for pointing this out. We agree with your comments. The changes are marked in red letters.
|
||
|
4. Response to Comments on the Quality of English Language |
||
|
Point 1: |
||
|
Response 1: The Quality of English Language in current version of our manuscript was improved by American professional Scientifics (in red). |
||
|
5. Additional clarifications |
||
|
The current version of our manuscript was critically revised and evaluated by several professional Scientifics. Changes are marked in red letters. Figure quality was improved. |
||
Reviewer 3 Report
Comments and Suggestions for Authors
Overall, I have concerns regarding the quality of this manuscript. There are several evident errors:
-
In the abstract, the statement "Cervical cancer (CC) is at the top of common cancer in incidence and mortality in females at worldwide" is problematic. This is a misleading formulation, as breast cancer is actually the most common cancer with the highest incidence and mortality among women, whereas CC is a cancer with relatively high—but not the highest—incidence and mortality in females.
-
The methodology section contains clear mistakes. For example, the official website of miRPathDB V2.0 [24] is no longer accessible. In addition, the CISTROME database is not a microRNA–protein target database but rather a transcription factor database, and therefore it cannot be used to query microRNAs targeting specific genes.
Given these issues, I have substantial concerns about the overall quality of the paper.
Author Response
|
1. Summary |
|
|
|
Thank you very much for taking the time to review this manuscript. Please find the detailed responses below and the corresponding revisions/corrections highlighted/in track changes in the re-submitted files.
|
||
|
2. Questions for General Evaluation |
Reviewer’s Evaluation |
Response and Revisions |
|
Does the introduction provide sufficient background and include all relevant references? |
Yes/Can be improved/Must be improved/Not applicable |
Not applicable |
|
Are all the cited references relevant to the research? |
Yes/Can be improved/Must be improved/Not applicable |
Not applicable |
|
Is the research design appropriate? |
Yes/Can be improved/Must be improved/Not applicable |
Not applicable |
|
Are the methods adequately described? |
Yes/Can be improved/Must be improved/Not applicable |
Not applicable |
|
Are the results clearly presented? |
Yes/Can be improved/Must be improved/Not applicable |
Not applicable |
|
Are the conclusions supported by the results? |
Yes/Can be improved/Must be improved/Not applicable |
Not applicable |
|
3. Point-by-point response to Comments and Suggestions for Authors |
||
|
Comments 1: In the abstract, the statement "Cervical cancer (CC) is at the top of common cancer in incidence and mortality in females at worldwide" is problematic. This is a misleading formulation, as breast cancer is actually the most common cancer with the highest incidence and mortality among women, whereas CC is a cancer with relatively high—but not the highest—incidence and mortality in females. |
||
|
Response 1: Thank you for pointing this out. We agree with this comment. The changes are marked in red letters: Cervical cancer (CC) ranks as the third common cancer……
Comments 2: The methodology section contains clear mistakes. For example, the official website of miRPathDB V2.0 [24] is no longer accessible. In addition, the CISTROME database is not a microRNA–protein target database but rather a transcription factor database, and therefore it cannot be used to query microRNAs targeting specific genes. Response 2: Thank you for pointing this out. We do not agree with this comment given that website of miRPathDB V2.0 was accessible when we the used it in July 2023. We send the complete table as supplementary file and evidence. Moreover, we never used CISTROME database for microRNA–protein target database as you mentioned. In the section 2.4, we described: For RUNX1, the target genes were identified using CISTROME database………
|
||
|
4. Response to Comments on the Quality of English Language |
||
|
Point 1: |
||
|
Response 1: We agree with this comment. Therefore, the Quality of English Language in current version of our manuscript was improved by American professional Scientifics (in red). |
||
|
5. Additional clarifications |
||
|
The current version of our manuscript was critically revised and evaluated by several professional Scientifics. Changes are marked in red letters. Figure quality was improved. |
||
Round 2
Reviewer 1 Report
Comments and Suggestions for Authors
Author Response
|
Response to Reviewer 1 Comments
|
||
|
1. Summary |
|
|
|
Thank you very much for taking the time to review this manuscript. Please find the detailed responses below and the corresponding revisions in track changes in the re-submitted files.
|
||
|
2. Questions for General Evaluation |
Reviewer’s Evaluation |
Response and Revisions |
|
Does the introduction provide sufficient background and include all relevant references? |
Yes |
|
|
Are all the cited references relevant to the research? |
Yes |
|
|
Is the research design appropriate? |
Can be improved |
|
|
Are the methods adequately described? |
Can be improved |
|
|
Are the results clearly presented? |
Can be improved |
|
|
Are the conclusions supported by the results? |
Can be improved |
|
|
3. Point-by-point response to Comments and Suggestions for Authors |
||
|
Comments 1:
|
||
|
Response 1: Thank you for the review report form. We agree with the comments provided. Therefore, we critically revised all manuscript, and some changes were performed (marked in red letter) to improved it. |
||
|
|
||
|
|
||
|
4. Response to Comments on the Quality of English Language |
||
|
Point 1: |
||
|
Response 1: The English quality was reviewed again by a native speaker. The changes are highlighted in red in the new version of the article. |
||
|
5. Additional clarifications |
||
|
The quality of the figures was improved in the Illustrator program considering 300 ppp for each one and exported to JPEG format. All figures were added to final version of our manuscript. In addition, we uploaded all independently figures to the system. |
||
Reviewer 3 Report
Comments and Suggestions for Authors
I am not sure whether the authors have noticed the serious bias in the data: in TCGA, the number of normal samples is only three, while in GSE86100 the normal and cancer samples are 6 versus 6. How can the results derived from such data be considered reliable?
Comments on the Quality of English Language
Not applicable
Author Response
|
Response to Reviewer 3 Comments
|
||
|
1. Summary |
|
|
|
Thank you very much for taking the time to review this manuscript. Please find the detailed responses below and the corresponding revisions/corrections highlighted/in track changes in the re-submitted files.
|
||
|
2. Questions for General Evaluation |
Reviewer’s Evaluation |
Response and Revisions |
|
Does the introduction provide sufficient background and include all relevant references? |
Not applicable |
|
|
Are all the cited references relevant to the research? |
Not applicable |
|
|
Is the research design appropriate? |
Not applicable |
|
|
Are the methods adequately described? |
Not applicable |
|
|
Are the results clearly presented? |
Not applicable |
|
|
Are the conclusions supported by the results? |
Not applicable |
|
|
3. Point-by-point response to Comments and Suggestions for Authors |
||
|
Comments 1: I am not sure whether the authors have noticed the serious bias in the data: in TCGA, the number of normal samples is only three, while in GSE86100 the normal and cancer samples are 6 versus 6. How can the results derived from such data be considered reliable?
|
||
|
Response 1: Thank you for pointing this out, however we respectfully disagree. The TCGA dataset is widely used even though it contains only 3 control samples. In addition, the ANOVA statistical test was applied, which indicates that there are significant differences between the groups. Similar results were obtained in the GSE86100 dataset, therefore, the results are consistent in two (independent) patient cohorts, indicating that these results are considered reliable and supporting our results.
|
||
|
4. Response to Comments on the Quality of English Language |
||
|
Point 1: |
||
|
The English quality was reviewed again by a native speaker. The changes are highlighted in red in the new version of the article. |
||
|
5. Additional clarifications |
||
|
The quality of the figures was improved in the Illustrator program considering 300 dpi for each one. |
||
Round 3
Reviewer 3 Report
Comments and Suggestions for Authors
OK